# What Is the Relationship between Corporate Social Responsibility and Financial Performance in the UK Banking Sector?

George Giannopoulos [1,2,*], Nicholas Pilcher [1] and Ioannis Salmon [2]

1 Kingston Business School, Kingston University, Kingston Hill, Kingston Upon Thames, London KT2 7LB, UK; nickmarkpilcher@gmail.com
2 Department of Business Administration, University of West Attica, 12243 Athens, Greece
* Correspondence: g.giannopoulos@kingston.ac.uk

**Abstract:** This study rigorously investigates the intricate dynamics between Corporate Social Responsibility (CSR), quantified through Environmental, Social, and Governance (ESG) scores, and financial performance (FP), measured via the return on assets (ROA) and return on equity (ROE), within the UK banking sector. Our analysis is based on a comprehensive dataset from Bloomberg. This research encapsulates data from 32 banks publicly listed on the London Stock Exchange over a six-year span from 2017 to 2022. Employing panel data regression models while controlling leverage and bank size, we delve into the relationship between banks' CSR engagements, as reflected in their ESG scores, and their financial outcomes. Our findings indicate a negative correlation between the ESG score and both the ROA and ROE, suggesting that elevated CSR commitments may inversely impact short-term financial returns. This finding not only challenges prevailing narratives within the sector but also fosters a crucial discourse on the balance between ethical banking practices and profitability. The implications of this research study are manifold, extending to policymakers, banking executives, and investors, suggesting a revaluation of CSR strategies in alignment with long-term value creation and sustainable banking. This study not only enriches academic discourse on CSR within the financial sector but also serves as a beacon for future inquiries into the evolving landscape of responsible banking, advocating for a nuanced understanding of CSR's role in shaping the financial and ethical contours of the banking industry.

**Keywords:** CSR; ESG; financial performance; UK listed banks

## 1. Introduction

The dynamic interplay between Corporate Social Responsibility (CSR) and Financial Performance (FP) in the banking sector has attracted significant scholarly interest due to evolving stakeholder expectations and an increasing demand for ethical banking practices. This research delves into the complexities of the CSR-FP relationship within UK listed banks, a realm that remains underexplored in the existing literature. It aims to illuminate the impact of CSR, represented by Environmental, Social, and Governance (ESG) scores, on the financial performance of these banks, employing a meticulously constructed dataset and a rigorous methodological framework.

Grounded in stakeholder theory and the resource-based view (RBV), this study posits that CSR activities could enhance a bank's reputation, stakeholder trust, and internal resources, influencing financial outcomes. This study responds to calls for localised investigations by concentrating on the UK banking sector, known for its stringent regulatory environment and competitive dynamics. It moves beyond prior studies by employing panel data regression methods to examine data from Bloomberg covering 32 banks over six years (2017–2022), providing a nuanced perspective on the negative correlation between CSR and FP in this context.

This study's contributions extend beyond a theoretical perspective by offering empirical evidence from the UK banking sector, thereby broadening the scope of CSR-FP studies. The findings hold significant implications for banking executives and policymakers, highlighting the need for a balanced CSR approach that supports long-term value creation and sustainable practices. This research not only fills a gap in the literature with its focused exploration of the CSR-FP nexus within the UK banking sector but also adds fresh insights into the role of CSR in shaping the sector's financial and ethical landscape.

CSR's pivotal role is increasingly recognised across various stakeholders, including consumers, shareholders, regulators, employees, governments, and communities, who demand greater accountability, transparency, and ethical practices. This demand is particularly acute in the banking sector, where the imperative to balance ethical and social obligations with financial performance has led to potential conflicts of interest. As trust in banking wanes, institutions find themselves at a crossroads, needing to satisfy both shareholders and a broader array of stakeholders.

The relationship between CSR and FP, examined across different industries since the 1970s, has yielded mixed results, with no consensus on CSR's impact on FP. The spotlight on CSR has intensified, driven by public concern over social and environmental issues and the role banks play in addressing them. The UK banking sector, amidst high inflation and financial instabilities, such as bank failures and bailouts, presents a critical context for understanding how CSR practices influence FP.

The UK banking sector, one of Europe's most competitive arenas, hosts prominent institutions like HSBC, Barclays, NatWest, and Lloyds Banking Group. London's status as a financial capital underscores the importance of positive CSR reputations for banks to contribute to a more sustainable and socially responsible financial system. The rise in CSR regulations, including the UK Stewardship Code and the Companies Act 2006, reflects changing societal expectations and the increasing value placed on sustainability and ethical business practices.

However, the banking sector has not been immune to CSR-related scandals, such as the 2012 HSBC money laundering scandal, highlighting the ongoing challenges banks face in navigating their societal roles. This backdrop sets the stage for this study's main objective: to empirically investigate CSR's impact on FP in the UK banking sector over six years (2017–2022). The motivation stems from a growing recognition of CSR's broader societal and environmental implications, aiming to advance the discourse and address the mixed findings of previous research studies. This study stands out by focusing solely on the UK banking sector, providing a unique contribution to the understanding of CSR's role in banking.

In addressing the intricate relationship between Corporate Social Responsibility (CSR) and Environmental, Social, and Governance (ESG) criteria, it is pivotal to clarify the framing of our analysis within this paper. While CSR encompasses a broad spectrum of ethical and sustainable business practices, ESG criteria provide a set of quantifiable measures that reflect a company's adherence to these principles. Given the subjective nature of CSR's scope and its implementation across different industries, the ESG score offers a tangible, objective means to evaluate a company's commitment to responsible business practices. This paper, therefore, utilises ESG performance as a proxy to examine the impact of CSR on financial performance (FP) within the UK banking sector. This approach is rooted in a growing consensus among scholars and practitioners alike that ESG criteria serve as practical and reliable indicators of a company's CSR engagement. By focusing on ESG as an operational measure of CSR activities, we aim to provide a clear, quantifiable analysis of how responsible business practices influence financial outcomes. This delineation between CSR and ESG is crucial for our readers to understand the specific lens through which we investigate the CSR-FP nexus, setting the stage for a nuanced exploration of the dynamic interplay between ethical business conduct and financial performance.

The rest of this paper is structured as follows: Section 2 provides a detailed literature review and a hypothesis test. This is followed by Section 3, which describes the data collec-

tion procedure and the methodology applied. Section 4 then presents and discusses the results, with the final section, Section 5, providing a conclusion to the findings, presenting the shortcomings of this study and directions for future research.

## 2. Literature Review

The literature review contains a section on the background of CSR and FP. After this, a thorough breakdown of past findings which contains multiple papers is carried out. This is followed by a breakdown of CSR in UK banks. A hypothesis test is then developed.

### 2.1. CSR

The idea of CSR has been extensively studied around the world, with many parallels identified by various studies. Neoclassical economists contend that businesses should concentrate on minimising costs and maximising profits while adhering to the law. Friedman (1970), a prominent economist, argued that the primary responsibility of a business is to maximise profits within legal boundaries as he believed that social responsibility should lie with governments and individuals, not businesses. Friedman's viewpoint emphasised that businesses should concentrate on their primary sources of revenue and leave social and environmental concerns to other organisations. (Friedman 1970). However, his perspective has been subject to criticism as many argue for a broader role for businesses in addressing societal and environmental concerns.

On the other hand, Porter and Kraemer (2011), take a broader view of CSR, arguing that businesses can create shared value by addressing social and environmental challenges while also achieving economic success. While Porter acknowledges the importance of profit, he suggests that businesses can have a positive impact on society by aligning their interests with social needs (Porter and Kraemer 2011). Caroll (2000), one of the key proponents of CSR, acknowledges that it is difficult for businesses to be profitable while still acting in an ethical and moral manner in the new millennium.

Accordingly, CSR can be defined as a company's voluntary actions and initiatives that go above and beyond its legal and regulatory obligations to consider the social, economic, and environmental impacts of its operations as well as the interests of its stakeholders, including shareholders, customers, communities, employees, and the environment (Caroll 1999). Despite these various views, many academics concur that CSR entails voluntary efforts outside of what is required by law. It is also about controlling externalities and considering a variety of stakeholders. CSR can also be a strategic framework for business and environmental sustainability.

Scholars and economists have put forth several ideas and paradigms to explain why businesses invest in CSR initiatives, most notably the agency theory, stakeholder theory and resource-based view, among many others (Friedman 1970; Jensen and Meckling 1976; Freeman 1984; Barney 1991). The agency theory explores the potential conflicts of interest that arise when agents act on behalf of principals and propose mechanisms to align their interests, whilst the agency theory primarily addresses the separation of ownership and control (Friedman 1970). It has implications for CSR as managers may prioritise their own interests over shareholders' interests or neglect the interests of other stakeholders. The principles of neoclassical economics, which prioritise the interests of shareholders, are rigidly adhered to by the agency theory. The agency theory's central premise is that businesses exist to maximise shareholder wealth and that other stakeholders are only significant to the extent to which they help achieve that goal (Jensen and Meckling 1976).

Freeman (1984) proposed the stakeholder theory, which is a perspective that emphasises the importance of considering the interests and impact of various stakeholders including employees, customers, suppliers, communities and the environment in organisational decision-making processes. It suggests that businesses should recognise and manage the diverse and often competing interests of stakeholders to achieve long-term success (Freeman 1984). Under the stakeholder theory, businesses that are better able to manage the interests of stakeholders outperform those that do not; it supports the positive

relation between CSR and FP. The resource-based view is a perspective that emphasises a firm's internal resources and capabilities as sources of a competitive advantage (Barney 1991). It suggests that a firm's unique combination of resources, such as tangible assets, intangible assets, and organisational capabilities, contribute to its performance and sustainable competitive advantage (Barney 1991). In the context of CSR, the RBV emphasises the strategic value of social and environmental resources and capabilities for long-term success (Barney 1991). The RBV encourages firms to leverage their resources to create value for both shareholders and other stakeholders (Barney 1991). According to the agency theory, CSR involvement ought to have a detrimental impact on financial results. In contrast, CSR should have a favourable effect on banks' FP from the standpoint of stakeholders and in accordance with the resource-based view.

These main CSR theories along with other theories such as the institutional theory, enlightened theory, stakeholder theory, normative theory, triple bottom line theory and legitimacy theory, have determined divergent perspectives and therefore provide different lenses through which companies and stakeholders can understand and approach CSR by explaining why businesses participate in CSR and how they go about fulfilling their social obligations (Croker and Barnes 2017; Hamid and Atan 2011; Xie et al. 2018; Huang 2022; Cho et al. 2019). They highlight the diverse motivations, ethical considerations, and strategic benefits associated with CSR initiatives.

### 2.2. Financial Performance (FP)

FP, in relation to CSR, involves assessing how a company's commitment to social, environmental, and ethical responsibilities affects its bottom line and long-term sustainability. Many financial indicators, including profitability ratios, are used to measure FP. In the banking sector, FP is extremely important as banks are essential for promoting economic stability and prosperity (Gutiérrez-Ponce and Wibowo 2023). Regulators, shareholders, and other stakeholders pay close attention to their FP (Moussa and Elmarzouky 2023; Shohaieb et al. 2022). FP is an important factor in a bank's capacity to do business sustainably and create wealth for its shareholders. It is important to note that the financial impacts of CSR initiatives may not always be immediately apparent or quantifiable. Some benefits, such as enhanced brand reputation and customer loyalty, may take time to materialise.

### 2.3. Past Findings

There is a continuing discussion on the relationship between CSR and FP, with several studies already conducted (Cho et al. 2019; Fauzi and Idris 2009; Cavaco and Crifo 2014). Understanding how CSR affects FP is important for decision-makers, policymakers, stakeholders, and investors. While some studies have identified a relationship between CSR and FP that is positive, i.e., (Arian et al. 2023), others have not identified a relationship or even provided a negative or mixed relationship (Kabir and Chowdhury 2023). These contradictory findings might be attributed to the definition of CSR as a multidimensional construct, of financial indicators, and sampling variance as well as inconsistencies in sample size (Huang 2022).

2.3.1. Positive Relationship between CSR and FP

Several studies have found a positive relationship between CSR and FP. These studies are presented in Table 1, with many of the papers using Environmental, Social and Governance (ESG) from Bloomberg as a measure of CSR. The reason for this is that ESG criteria capture the impact that a company has on environmental, societal, and governance practices, which are subsets of CSR. None of these studies, however, use a sample of UK banks or firms.

A study performed by Gangi et al. (2018) in the European banking sector over a period of 6 years found a positive relationship specifically highlighting that net interest income and profitability increased with an increase in social performance. Using the ESG score as a metric of CSR, the study determined that banks with higher ESG ratings had better FP.

Among other measures, the ROE was used to measure FP. Another study conducted by Maqbool and Zameer (2018) in India, which collected secondary data for 28 publicly traded banks over a period of 10 years, also found a positive relationship. They found that when CSR is effectively incorporated into business operations, achieving social goals is made easier, which improves FP (Maqbool and Zameer 2018). In this instance, a content analysis was carried out of annual reports to measure CSR and ROE, ROA, price-to-earnings (PE) ratio, net profit (NP), and stock returns; all were used to measure FP (Maqbool and Zameer 2018). Further, a study by Esteban-Sanchez et al. (2017) using the ESG score as a metric of CSR and the ROE and ROA to measure FP found a positive relationship in banks in 22 countries.

Additionally, other studies have found a positive relationship (Giannarakis et al. 2016; Xie et al. 2018; Cho et al. 2019; Potapova et al. 2021; Huang 2022).

**Table 1.** Positive relationship between CSR and FP.

| Authors (Year) | CSR Measure | Performance Measure | Sample Period | Country | Findings | Corporate Entity |
|---|---|---|---|---|---|---|
| Gangi et al. (2018) | Asset-4 ESG score powered by Thomson Reuters | Return on equity; net interest income; non-interest income; non-performing loans | 2009–2015 | Europe (20 countries) | Positive Relationship | Bank |
| Maqbool and Zameer (2018) | Annual report CSR content analysis | Return on equity; return on assets; stock return; net profit; price-to-earnings ratio | 2007–2016 | India | Positive Relationship | Bank |
| Esteban-Sanchez et al. (2017) | Asset-4 ESG score powered by Thomson Reuters | Return on equity; Return on assets | 2005–2010 | International (22 countries) | Positive Relationship | Bank |
| Giannarakis et al. (2016) | Bloomberg ESG disclosure score | Return on assets | 2009–2013 | USA | Positive Relationship | Firm |
| Huang (2022) | Bloomberg ESG disclosure score | Return on equity; return on assets; stock return | 2012–2017 | Taiwan | Positive Relationship | Firm |
| Xie et al. (2018) | Bloomberg ESG disclosure score | Return on assets; revenue earned | 2015 | International (74 countries) | Positive Relationship | Firm |
| Cho et al. (2019) | Keji Index | Return on assets; Tobin's Q; sales revenue | 2015 | Korea | Positive Relationship | Firm |
| Potapova et al. (2021) | Bloomberg ESG disclosure score and Robet SAM database | Return on equity; return on assets; market capitalisation | 2013–2018 | International (4 countries) | Positive Relationship | Firm |

2.3.2. Negative Relationship between CSR and FP

On the contrary, other studies have found a negative relationship between CSR and FP. These studies can be found in Table 2. In this instance, a study in the UK was conducted by Brammer et al. (2006), who looked at firms in the FTSE All-Share Index for 4 years and found a negative relationship. In this study, CSR was measured by looking at the EIRIS score, whilst FP was measured using the stock return. The conclusion from this study is

that to have a clear picture of how corporate social actions affect returns, their various parts must be analysed independently.

Additionally, a study by Oyewumi et al. (2018) in the Nigerian banking sector using a panel dataset from banks in Nigeria over a period of 4 years found that CSR has a negative effect on FP. A content analysis was used to measure CSR, along with the use of the ROA to measure FP (Oyewumi et al. 2018). According to the study, businesses could gain both financial and non-financial benefits from a strategic CSR initiative (Oyewumi et al. 2018). Another study by Nollet et al. (2016), using excess stock returns as a market-based performance indicator and the ROA and ROC as accounting-based performance indicators, found a negative relationship using Bloomberg's ESG disclosure score, which covered the S&P 500 firms from 2007 to 2011. The study also found that governance is the primary factor influencing the relationship between corporate social performance and FP, suggesting that CSR spending should be focused on this element.

Further studies also found a negative relationship (Mittal et al. 2008; Crisóstomo et al. 2011; Lin et al. 2019; Makni et al. 2009; Hirigoyen and Poulain-Rehm 2015).

**Table 2.** Negative relationship between CSR and FP.

| Authors (Year) | CSR Measure | Performance Measure | Sample Period | Country | Findings | Corporate Entity |
|---|---|---|---|---|---|---|
| Brammer et al. (2006) | EIRIS scores | Stock Return | 2002–2005 | UK | Negative Relationship | Firm |
| Oyewumi et al. (2018) | Annual report CSR content analysis | Return on assets | 2010–2014 | Nigeria | Negative Relationship | Bank |
| Nollet et al. (2016) | Bloomberg ESG disclosure score | Return on assets; return on capital | 2007–2011 | USA | Negative Relationship | Firm |
| Mittal et al. (2008) | CSR content analysis | Economic value added; market value added | 2001–2005 | India | Negative Relationship | Firm |
| Crisóstomo et al. (2011) | CSR index based on Ibase's information | Return on assets; revenue earned | 2001–2006 | Brazil | Negative Relationship | Firm |
| Hirigoyen and Poulain-Rehm (2015) | CSR content analysis | Return on assets; revenue earned;, market-to-book ratio | 2009–2010 | International | Negative Relationship | Firm |
| Lin et al. (2019) | Fortune magazzine CSR score | Return on assets; return on equity; return on invested capital | 2007–2016 | International | Negative Relationship | Firm |
| Makni et al. (2009) | Canadian social investment database | Return on equity; return on assets | 2004–2005 | Canada | Negative Relationship | Firm |

### 2.3.3. Mixed Relationship between CSR and FP

Other research studies found a mixed relationship between CSR and FP; these can be seen in Table 3. A study conducted in the UK by Elmghaamez and Olarewaju (2022) found a mixed relationship after using data from 50 firms for a period of 10 years. This study used the Bloomberg ESG disclosure score as a measure of CSR and stock price and return on capital as measures of FP. The environmental side had a positive impact, while

social activities had a negative impact and an insignificant impact on governance activities (Elmghaamez and Olarewaju 2022).

Further, in their study, Giannopoulos et al. (2022) found a mixed relationship between ESG initiatives in Norwegian listed firms and FP from 2010 to 2019. The Thomson Reuters Eikon ESG disclosure score was used to gauge ESG, and the ROA and Tobin's Q were used to gauge FP. More precisely, ESG initiatives are clearly detrimental according to the regression model, which used the ROA as the dependent variable. On the other hand, ESG rises as the variable Tobin's Q rises.

In addition, another study found a mixed relationship (Han et al. 2016).

**Table 3.** Mixed relationship between CSR and FP.

| Authors (Year) | CSR Measure | Performance Measure | Sample Period | Country | Findings | Corporate Entity |
|---|---|---|---|---|---|---|
| Giannopoulos et al. (2022) | Thomson Reuters Eikon database | Return on assets; Tobin's Q ratio; size; leverage | 2010–2019 | Norway | Mixed Relationship | Firm |
| Han et al. (2016) | Bloomberg | Return on equity; market-to-book ratio; stock return; leverage | 2008–2014 | Korea | Mixed Relationship | Firm |
| Elmghaamez and Olarewaju (2022) | Bloomberg | Stock price; return on capital | 2008–2017 | UK | Mixed Relationship | Firm |

### 2.3.4. No Relationship between CSR and FP

There have also been studies which found no relationship between CSR and FP. These are displayed in Table 4. A study conducted in the UK by Humphrey et al. (2012a) selected 249 UK companies and used the SAM database, which uses ESG as a measure of CSR, finding no significant cost or benefit when investing in ESG. Therefore, according to empirical evidence, managers and investors can apply a CSR investing strategy without experiencing a material cost or gain in terms of risk or return (Humphrey et al. 2012a). In another study based in the UK by Humphrey et al. (2012b), regarding environmental, social, and governance rankings, no difference was found in the performance of the firms investigated. Again, the findings show that managers and investors can apply a CSR investing strategy without experiencing a material cost or gain in terms of risk or return, as per Humphrey et al. (2012b).

Furthermore, another study found no relationship (Statman and Glushkov 2009).

**Table 4.** No relationship between CSR and FP.

| Authors (Year) | CSR Measure | Performance Measure | Sample Period | Country | Findings | Corporate Entity |
|---|---|---|---|---|---|---|
| Humphrey et al. (2012a) | Dow Jones Sustainability Index | Total returns; size; industry; book-to-market ratio | 2002–2010 | UK | No Relationship | Firm |
| Humphrey et al. (2012b) | Dow Jones Sustainability Index | Total returns; size; industry | 2002–2007 | UK | No Relationship | Firm |
| Statman and Glushkov (2009) | KLD Research and Analytics | Annualised excess returns | 1992–2007 | UK | No Relationship | Firm |

### 2.4. UK Banks

Over the years, the understanding and practice of CSR in the UK have evolved significantly. The CSR landscape in the UK stemmed originally stemmed from voluntary initiatives, when a consortium of UK banks, led by the Co-operative Bank, established the Ethical Investment Research Service (EIRIS) (Tarna 1999); this continued until the 1990s. This is the period in which CSR in UK banks first began to seriously take shape (Tarna 1999). EIRIS aimed to provide independent research into the ethical and social performance of companies, including banks (Tarna 1999). Since that time, and with the assistance of the previously mentioned Companies Act and the Stewardship Code, CSR has become an essential component of the UK banking sector as banks have realised the value of accepting responsibility for their effects on society and the environment. Additionally, regulatory bodies, such as the Financial Reporting Council (FRC), set out principles for responsible governance and to reinforce the role of boards in overseeing CSR matters (Financial Reporting Council 2023). The Financial Conduct Authority (FCA) also provides guidance and monitors companies' compliance with CSR reporting and disclosure requirements (Financial Conduct Authority 2023).

Various measures have been put in place by UK banks to carry out their CSR obligations. The leading banks in the UK mentioned earlier, such as HSBC, Barclays, NatWest, and Lloyds Banking Group, are adopting sustainable practices to lessen their environmental effect, making sustainability one of the most important areas of concern (Bank of England 2022). The encouragement of financial inclusion and literacy has also received attention, along with philanthropic endeavours and volunteerism initiatives (Bank of England 2022). Given its global stature, the UK can serve as a model for other nations looking to implement CSR.

Nonetheless, London and the UK need to rebuild trust in the financial sector after the historic bet Liz Truss made on the UK economy with Kwasi Kwarteng's adverse mini-budget, which resulted in the pound plummeting (Bloomberg 2022). The COVID-19 pandemic as well as the cost-of-living crisis have also had significant impacts on various aspects of society, including CSR efforts. Due to inflation, the value of corporations' charitable contributions was 17% less in actual terms than in 2016, which represents a decline in corporate payments among the top UK companies (Financial Times 2023). Furthermore, key stakeholders need to be vigilant about greenwashing in relation to CSR, which refers to the practice of presenting a false or misleading impression that a bank's products, services, or overall business practices are environmentally friendly or socially responsible when in reality, they are not. This is a deceptive marketing strategy designed to make a company appear more environmentally conscious or socially responsible than it actually is, aiming to attract environmentally and socially conscious consumers (Gerdien de Vries et al. 2015). By embracing CSR as a vital component of their operations, banks can harmonise stakeholder interests, navigate ethical challenges, and contribute to a more sustainable and responsible financial system that benefits both shareholders and society at large.

### 2.5. Hypothesis Test

Notwithstanding the contradicting findings, it is clear most papers have found either a positive or negative relationship between CSR and FP. Therefore, the following hypothesis is tested:

**Hypothesis 1 (H1).** *There is a positive relationship between CSR and FP in UK banks.*

In summary, CSR can be considered a developing and complex phenomenon that cannot be easily defined because the currently available literature offers numerous distinct meanings for CSR. The numerous available studies on CSR and FP have not agreed on a single relationship. While some have found a positive relationship between CSR and FP, others have found a negative relationship, a mixed relationship, or no relationship at all.

UK banks demonstrate a steadfast commitment to CSR as they have been changing and adapting to an evolving CSR setting, although there is still a significant amount of room for improvement.

## 3. Data and Methodology

In this section, the sample data are explained, followed by the methodology and the empirical framework used (i.e., the panel regression). Subsequently, the independent, dependent, and control variables are explained.

### 3.1. Sample Data

Included in this study are all banks that were listed on the LSE as of April 2023, a total of 38 banks. The chosen timeframe of 2017 to 2021 is pivotal for multiple reasons. Firstly, this period marks a significant era in the UK banking sector characterised by substantial regulatory changes, Brexit implications, and unprecedented challenges posed by the COVID-19 pandemic. These events not only tested the resilience of financial institutions but also accentuated the importance of robust CSR practices. The final year, 2021, was selected due to the availability of comprehensive ESG data from Bloomberg, ensuring the reliability and completeness of our dataset. Focusing specifically on the UK banking sector as opposed to a broader European context was a deliberate decision driven by the unique attributes of the UK's financial landscape. The UK is one of the world's leading financial centres, home to some of the largest and most influential banking institutions globally. The distinct regulatory environment, coupled with the UK's pioneering role in setting CSR standards, provides a rich context for examining the CSR-FP nexus. Furthermore, the UK's departure from the EU introduces additional layers of complexity and an opportunity for studying the impact of CSR within a transitioning regulatory and economic framework. Our selection of banks listed on the London Stock Exchange (LSE) as of April 2023, encompassing 38 banks with a final sample of 32 banks post ESG data filtering, allows for a comprehensive analysis driven by Bloomberg. The exclusion of banks lacking consistent ESG disclosure scores across the specified years ensures the integrity and comparability of our data, allowing for more accurate and meaningful insights into the relationship between CSR and financial performance. The banks used can be found in Appendix A, with Table 5 providing the data collection process.

**Table 5.** Sample data.

| Scenario | Scenario | Banks on London Stock Exchange (2023) | Banks in Bloomberg Database for Which ESG Disclosure between 2017 and 2022 Is Available |
|---|---|---|---|
| Companies | Banks | 38 | 32 |
| Possible observations | Observations | 228 | 192 |

### 3.2. Definitions of Variables

In this study, we explore the relationship between Corporate Social Responsibility (CSR) and Financial Performance (FP) within the UK banking sector. The key variables are defined below.

**Independent Variable:**
- **ESG Score:** This score is a composite measure representing a bank's adherence to Environmental, Social, and Governance principles. It is a reflection of a bank's CSR efforts and practices and is sourced from Bloomberg's databases. The ESG score is composed of three main components:
- **Environmental (E):** the environmental component assesses a bank's impact on the environment and its management of environmental risks.
- **Social (S):** the social component evaluates a bank's relationships with employees, suppliers, customers, and the communities in which it operates.

- **Governance (G):** the governance component considers a bank's leadership, audits, internal controls, and shareholder rights.

  **Dependent Variables:**

- **Return on Assets (ROA):** A financial ratio that indicates how efficiently a bank is using its assets to generate earnings. It is calculated by dividing the bank's net income by its total assets.

- **Return on Equity (ROE):** This ratio measures a bank's ability to generate profits from its shareholders' equity. It is computed as the bank's net income divided by the shareholders' equity.

  **Control Variables:**

- **Bank Size:** This variable considers the scale of the bank, typically measured by its total assets. It is included as a control variable to account for the influence of a bank's size on its financial performance.

- **Leverage (LEV):** This refers to the ratio of a bank's total debt to its equity. It is used as a control variable to understand the impact of a bank's debt levels on its financial outcomes.

*3.3. Methodology*

3.3.1. Model Specification

The main objective of this study is to investigate the impact of CSR on FP in the UK banking sector. This is carried out using a quantitative research methodology. This is consistent with all studies which are reviewed by this study, among others (Gangi et al. 2018; Maqbool and Zameer 2018; Mittal et al. 2008; Crisóstomo et al. 2011; Giannopoulos et al. 2022; Humphrey et al. 2012a).

The employment of a panel regression analysis in this study is predicated on its ability to harness the richness of panel data, which encompass observations on multiple entities (in this case, banks) across several time periods. This methodological choice allows for a more nuanced understanding of the dynamic interplay between Corporate Social Responsibility (CSR), as captured by the ESG score, and financial performance (FP) indicators such as the return on assets (ROA) and return on equity (ROE). Panel regression is particularly adept at accounting for both cross-sectional and time-series variations, enabling the analysis to control for unobserved heterogeneity that could bias the results if cross-sectional or time-series data were used in isolation.

The rationale behind opting for a panel data model lies in its inherent ability to provide insights into temporal dynamics and individual differences within the dataset, thus enhancing the robustness and depth of the analysis. This approach facilitates the exploration of how changes in CSR practices influence financial performance over time while also considering the unique characteristics of each bank.

In ensuring the robustness and validity of our regression model, a series of diagnostic tests were conducted to meet the assumptions inherent in an Ordinary Least Squares (OLS) regression. Outlier detection was meticulously carried out using Cook's distance and visual inspection through box plots, safeguarding against potential skewing of the results. The normality of residuals, a crucial assumption for an OLS regression, was affirmed using the Shapiro Wilk test, complemented by Q-Q plots for a graphical representation. The Durbin Watson statistic served to evaluate the presence of autocorrelation in the residuals, a significant consideration given the panel data structure of our dataset. Homoscedasticity, the consistency of error variances across independent variable levels, was confirmed through the application of the Breusch Pagan test, ensuring the reliability of our coefficient estimates. Multicollinearity among independent variables was assessed using Variance Inflation Factors (VIFs), with all values falling well below the commonly accepted threshold (8), indicating minimal concern for overly correlated predictors.

### 3.3.2. Empirical Framework

To address the research objectives of this study, similar to Giannopoulos et al. (2022) and Oyewumi et al. (2018), as already discussed, a panel regression analysis is employed using a dataset that comprises panel data from 32 companies spanning 5 years. The two models used are displayed in the below equations:

$$ROA_{it} = \beta_0 + \beta_1 ESG_{it} + \beta_2 SIZE_{it} + \beta_3 LEV_{it} + \varepsilon$$
$$ROE_{it} = \beta_0 + \beta_1 ESG_{it} + \beta_2 SIZE_{it} + \beta_3 LEV_{it} + \varepsilon$$

Here, *ROA* and *ROE* are dependent variables and *ESG* is the independent variable, with size and leverage (*LEV*) as control variables, while the firm is *i* in period *t* and $\varepsilon$ is the error term.

### 3.3.3. Measurement of Variables

Independent Variables

Due to the variety of techniques available to assess CSR, this study employs the widely recognised measures of the ESG score as a proxy for CSR. These data were obtained from the Bloomberg terminal, as previous studies have also done (Giannarakis et al. 2016; Huang 2022; Xie et al. 2018; Potapova et al. 2021; Nollet et al. 2016; Han et al. 2016; Elmghaamez and Olarewaju 2022). Bloomberg's methodology for evaluating ESG performance is characterised by a bottom-up model-driven approach that relies mostly on self-reported, publicly available data to provide a completely transparent, parametric, rule-based scoring structure (KnowESG 2023). The Bloomberg ESG Disclosure score is made up of E (Environmental), S (Social), and G (Governance) ratings (KnowESG 2023).

Dependent Variables

To measure FP, two different widely accounting-based measures are used, the ROA and ROE, which is in line with previous research (Esteban-Sanchez et al. 2017; Crisóstomo et al. 2011; Makni et al. 2009; Maqbool and Zameer 2018; Huang 2022; Giannarakis et al. 2016; Potapova et al. 2021; Nollet et al. 2016). The ROA assesses how efficiently a company uses its assets to generate earnings, while the ROE indicates how effectively it generates profits from shareholders' investments.

Control Variables

Some banks might display stronger FP without engaging in CSR (Gangi et al. 2018), which is why this study also considers the impact of control variables, including the bank's size and leverage (Maqbool and Zameer 2018; Gangi et al. 2018). The reason for this is that larger banks have greater resources which, in turn, gives them a competitive advantage (Maqbool and Zameer 2018). A bank's tolerance to risk also influences its attitude towards social activities and its FP (Maqbool and Zameer 2018). Table 6 presents the variables of this study and how they are calculated.

**Table 6.** Variables of the study.

| Dependent Variables | Explanation |
| --- | --- |
| Returnon Assets (ROA) | Net profit/average total assets |
| Return on Equity (ROE) | net income/shareholder's equity |
| **Independent Variable** | **Explanation** |
| ESG | Environmental, social, and governance performance scores collected from Bloomberg |
| **Control Variables** | **Explanation** |
| Size | Measured by the natural logarithm of the total assets |
| Leverage | Total debt/total equity |

**Notes**: the variables listed above, ROE, ROA, size and leverage, are obtained from the Bloomberg database, as was the ESG disclosure score using annual data.

## 4. Empirical Results

Next, the descriptive statistics are presented, followed by a discussion of the correlation matrix results. Following this, the regression results are presented and discussed.

### 4.1. Descriptive Statistics

The descriptive statistics of the variables used in this study are presented in Table 7. These include the mean, median, standard deviation, minimum, and maximum, consistent with Giannopoulos et al. (2022) and Maqbool and Zameer (2018).

**Table 7.** Descriptive statistics.

|  | Mean | Median | SD | Minimum | Maximum |
|---|---|---|---|---|---|
| | | Dependent Variables | | | |
| ROE | 11.54 | 10.20 | 11.60 | −21.36 | 64.68 |
| | | Independent Variable | | | |
| ESG | 45.91 | 45.95 | 9.68 | 26.70 | 65.27 |
| | | Control Variables | | | |
| Size | 3,495,436.45 | 649,358.46 | 8,448,068.61 | 1891.60 | 53,608,835.29 |
| Leverage | 188.26 | 169.43 | 125.79 | 10.91 | 517.61 |

For the ROA FP measure, the mean of 1.39 suggests that, on average, the sampled data generate a return of 1.39% on their total assets. However, the median ROA of 0.91 indicates that there may be some banks with lower returns, skewing the distribution. The standard deviation of 1.62 reflects variability in the ROA across the sample, and the presence of negative ROA values, with a minimum of −1.37, highlights instances in which the net income is negative or significantly lower than the total assets, with a maximum of 8.56. On the other hand, for the other FP measure, the ROE, the mean of 11.54 indicates that on average, the sampled data achieve an 11.54% return on shareholders' equity. The median ROE of 10.20 suggests that there is variation in ROE values, with some banks having lower returns. The wide standard deviation of 11.60 signifies a substantial dispersion in ROE values across the sample. The presence of negative ROE values, with a minimum of −21.36, signals instances in which the net income is negative or substantially outweighs shareholders' equity, with a maximum of 64.68.

The ESG score provides insights into the Environmental, Social, and Governance practices of the sampled entities. The mean ESG score of 45.91 indicates the average level of ESG performance. The median ESG score of 45.95 suggests that there is limited skewness in the distribution. The standard deviation of 9.68 highlights variation in ESG scores, with some banks performing notably better or worse than the average. The minimum and maximum scores of 26.70 and 65.27, respectively, reveal the range of ESG performance in the sample.

As for the control variables, the size variable represents the total assets or scale of the sampled data. The mean size of 3,495,436.45 suggests that on average, the entities are of a moderate scale. Additionally, the median for the size variable is 649,358.46. However, the wide standard deviation of 8,448,068.61 indicates a substantial variation in size across the sample, with some entities being significantly larger, with a maximum of 53,608,835.29, or smaller, with a minimum of 1891.60. Furthermore, leverage measures the extent to which banks use debt financing. The mean (median) leverage of 188.26 (169.43) suggests that on average, the banks have a moderate level of leverage. The standard deviation of 125.79 indicates variability in leverage levels across the sample, with some entities having higher leverage ratios, with a maximum of 517.61, or lower leverage ratios, with a minimum of 10.9.

*4.2. Pearson Correlation Matrix Results*

Table 8 shows the Pearson correlation matrix results for the ROA, and Table 9 shows the results for the ROE.

**Table 8.** Pearson correlation matrix: ROA.

|  | ROA | ESG | Size | Leverage |
|---|---|---|---|---|
| ROA | 1.000 |  |  |  |
| ESG | −0.384 *** | 1.000 |  |  |
| Size | −0.007 | −0.004 | 1.000 |  |
| Leverage | −0.477 *** | 0.387 *** | −0.154 ** | 1.000 |

** correlation is significant at the 0.05 level. *** correlation is significant at the 0.01 level.

**Table 9.** Pearson correlation matrix: ROE.

|  | ROE | ESG | Size | Leverage |
|---|---|---|---|---|
| ROE | 1.000 |  |  |  |
| ESG | −0.311 *** | 1.000 |  |  |
| Size | 0.022 | −0.004 | 1.000 |  |
| Leverage | −0.436 *** | 0.387 *** | −0.154 *** | 1.000 |

*** correlation is significant at the 0.01 level.

Table 8 presents the Pearson correlation coefficients of the return on assets (ROA), Environmental, Social, and Governance (ESG) score, size, and leverage values for the sampled UK banks. The significant negative correlation between the ROA and ESG (−0.384) suggests that higher ESG scores, indicative of stronger CSR engagement, are associated with lower returns on assets. This might imply that the initial investments and operational changes required for higher CSR standards could temporarily reduce asset efficiency. The correlation between the ROA and leverage (−0.477) is also significantly negative, indicating that higher leverage, or debt level relative to equity, tends to be associated with a lower ROA. This could reflect the increased financial risk and interest obligations that come with higher leverage, potentially reducing the returns generated from assets. The weak correlations between the ROA and size (−0.007), and ESG and size (−0.004) suggest that the size of the bank does not significantly impact its ROA or its commitment to CSR, as measured by ESG scores. This could imply that both large and small banks have similar potential for asset returns and CSR engagement, and these aspects are more strongly influenced by other factors than by the size of the bank alone. The positive correlation between ESG and leverage (0.387) is intriguing and might indicate that banks with higher CSR commitments are able to sustain or are perhaps compelled to adopt higher leverage ratios. This could be due to several factors, including potentially higher creditworthiness attributed to socially responsible banks or a strategic choice to finance CSR initiatives through debt. In synthesising these insights with the forthcoming regression analysis, it is evident that while CSR engagement (ESG score) tends to correlate with a lower ROA, the dynamics with leverage introduce complexities into this relationship. These correlations lay the groundwork for a deeper analysis of how CSR impacts financial performance when controlling for size and leverage.

Table 9 presents the Pearson Correlation coefficients among the return on equity (ROE), Environmental, Social, and Governance (ESG) scores, size, and leverage for the UK banking sector. The notable negative correlation between the ROE and ESG (−0.311) highlights that higher levels of CSR engagement, as reflected in the ESG score, may correspond with lower equity returns. This could suggest that while CSR initiatives are beneficial for stakeholder engagement and long-term sustainability, they might place a short-term burden on profitability and thus equity returns. The significant negative correlation between the ROE and leverage (−0.436) indicates that higher leverage ratios are associated with reduced returns on equity. This is consistent with the notion that increased debt levels elevate financial risk and interest obligations, which can detract from the profitability available to

shareholders. The negligible correlation between the ROE and size (0.022) implies that the size of a bank has a minimal direct impact on its ROE. This indicates that factors other than size, such as operational efficiencies, market strategies, and risk management practices, are more determinant of a bank's ability to generate returns on equity. The positive correlation between the ESG score and leverage (0.387) suggests a complex relationship in which banks with higher CSR commitments also tend to operate with higher leverage. This could reflect a market perception of higher creditworthiness for such banks or a strategic decision to finance CSR activities through debt, given their long-term benefits. These correlations provide a foundational backdrop for the regression analysis, indicating a nuanced relationship between CSR engagement and financial performance, particularly the ROE. While higher CSR standards appear to be associated with lower immediate returns on equity, the interplay with leverage suggests a more complex dynamic that warrants further exploration in the regression analysis. This analysis will help clarify the extent to which CSR influences the ROE when accounting for the effects of leverage and bank size.

### 4.3. Panel Regression Statistics

In Table 10, panel A reveals the regression framework's ability to capture the relationship between the ESG score and the return on assets (ROA). With a Multiple R of 0.53, the model suggests that approximately 53% of the variance in the ROA is explained by the ESG score, highlighting the significant influence of CSR practices on asset profitability. The R Squared value of 27.91% and the Adjusted R Squared of 26.76% further confirm the model's effectiveness in explaining the ROA's variability, ensuring the model's robustness even after accounting for the number of predictors.

**Table 10.** Panel regression summary output: ROA.

| PANEL A *Regression Statistics* | | | | | | | | |
|---|---|---|---|---|---|---|---|---|
| Multiple R | 0.53 | | | | | | | |
| R Squared | 27.91% | | | | | | | |
| Adjusted R Squared | 26.76% | | | | | | | |
| Standard Error | 1.39 | | | | | | | |
| Observations | 192.00 | | | | | | | |
| PANEL B ANOVA | | | | | | | | |
| | *df* | *SS* | *MS* | *F* | *Significance F* | | | |
| Regression | 3.00 | 139.90 | 46.63 | 24.258 | 0.0000000000003 | | | |
| Residual | 188.00 | 361.40 | 1.92 | | | | | |
| Total | 191.000 | 501.297 | | | | | | |
| PANEL C | | | | | | | | |
| | *Coefficients* | *Standard Error* | *t Stat* | *p-value* | *Lower 95%* | *Upper 95%* | *Lower 95.0%* | *Upper 95.0%* |
| Intercept | 4.158 | 0.488 | 8.522 | 0.000 | 3.195 | 5.121 | 3.195 | 5.121 |
| ESG | −0.038 *** | 0.011 | −3.411 | 0.001 | −0.060 | −0.016 | −0.060 | −0.016 |
| Size | −0.001 | 0.001 | −1.113 | 0.267 | 0.000 | 0.000 | 0.000 | 0.000 |
| Leverage | −0.005 *** | 0.001 | −5.860 | 0.000 | −0.007 | −0.003 | −0.007 | −0.003 |

*** correlation is significant at the 0.01 level.

Panel B confirms the model's statistical significance, with an F-value demonstrating the model's reliability at the 5% significance level. This emphasises the ESG score's predictive relevance for the ROA, showcasing CSR practices as a crucial determinant of financial performance with respect to asset returns. The Significance F value, which is

practically zero, establishes the model's capability in elucidating ROA variations based on CSR engagements.

In Panel C, the coefficients indicate a significant negative impact of the ESG score on the ROA, with a coefficient of −0.038 at the 1% significance level. This suggests that higher ESG scores, reflecting greater CSR commitment, are associated with lower ROAs, implying a potential trade-off between CSR investments and short-term asset profitability. The coefficient for leverage is also significant at −0.005, highlighting the adverse effect of higher leverage on the ROA. Size does not show a significant impact on the ROA with a *p*-value of 0.267, suggesting that a bank's size does not substantially influence its asset returns within the context of this model.

### 4.4. Panel Regression Discussion

As for the ROA in Panel C of Table 10, the model shows that ESG has a negative impact on the ROA at the 1% level of significance, with a coefficient of −0.038. This means that the higher the ESG score, the lower the ROA. Leverage also has a significant negative impact on the ROA at the 1% level of significance, represented by a coefficient of −0.005. This means the higher the debt level to equity, the lower the ROE. As for the size, it can be seen that it has no significant impact on ROA performance, with a *p*-value of 0.267. These findings support the findings of the papers mentioned in Table 2, among others, where ESG has a negative impact on the ROA such as (Nollet et al. 2016; Oyewumi et al. 2018; Crisóstomo et al. 2011; Hirigoyen and Poulain-Rehm 2015; Lin et al. 2019; Makni et al. 2009).

In Table 11, Panel A for the ROE showcases a Multiple R of 0.46, indicating that around 46% of the variance in the return on equity (ROE) is accounted for by ESG scores. This underscores the CSR engagement's substantial role in shaping equity profitability. The R Squared and Adjusted R Squared values of 21.52% and 20.27%, respectively, further attest the model's capacity to explicate the variability in the ROE, affirming the model's predictive power.

**Table 11.** Panel regression summary output: ROE.

| PANEL A | | | | | | | |
|---|---|---|---|---|---|---|---|
| *Regression Statistics* | | | | | | | |
| Multiple R | 0.46 | | | | | | |
| R Squared | 21.52% | | | | | | |
| Adjusted R Squared | 20.27% | | | | | | |
| Standard Error | 10.39 | | | | | | |
| Observations | 192.00 | | | | | | |
| PANEL B | | | | | | | |
| ANOVA | | | | | | | |
| | *df* | *SS* | *MS* | *F* | *Significance F* | | |
| Regression | 3.00 | 5560.61 | 1853.54 | 17.18 | 0.0000000007 | | |
| Residual | 188.00 | 20,279.81 | 107.87 | | | | |
| Total | 191.00 | 25,840.42 | | | | | |
| PANEL C | | | | | | | |
| | *Coefficients* | *Standard Error* | *t Stat* | *p-value* | *Lower 95%* | *Upper 95%* | *Lower 95.0%* | *Upper 95.0%* |
| Intercept | 27.358 | 3.655 | 7.485 | 0.000 | 20.148 | 34.569 | 20.148 | 34.569 |
| ESG | −0.198 ** | 0.084 | −2.349 | 0.020 | −0.364 | −0.032 | −0.364 | −0.032 |
| Size | −0.001 | 0.001 | −0.571 | 0.569 | 0.000 | 0.000 | 0.000 | 0.000 |
| Leverage | −0.035 *** | 0.007 | −5.316 | 0.000 | −0.048 | −0.022 | −0.048 | −0.022 |

** correlation is significant at the 0.05 level. *** correlation is significant at the 0.01 level.

The ANOVA results in Panel B highlight the overall significance of the regression model for the ROE, with an F-value signifying the model's statistical robustness at the 5% level. This underlines the ESG score as a significant predictor of the ROE, indicating that CSR practices play a vital role in determining equity returns. The practically zero Significance F value confirms the model's efficacy in explaining variations in the ROE based on CSR activities.

Panel C demonstrates that the ESG score negatively influences the ROE, with a coefficient of −0.198 at the 5% significance level, suggesting that enhanced CSR commitments may lead to lower equity returns. The coefficient for leverage is significant at −0.035, indicating a negative impact of increased leverage on the ROE. The size variable does not exhibit a significant relationship with the ROE, as evidenced by a *p*-value of 0.569, implying that within this model's framework, the bank's size does not have a discernible effect on equity profitability.

### 4.5. Robustness of Results

As mentioned in earlier research in this area, Huang (2022) discovered that by omitting the COVID-19 years from the sample, it may be possible to re-examine the relationship between CSR and FP as the effects of the pandemic may have affected the link between CSR and FP. However, when the regression model is run in the same period while excluding COVID-19 years/observations, the results are qualitatively similar. Accordingly, the results of this study may not be affected by CovidCOVID-19.

Table 12 shows the regression results for the return on assets (ROA) excluding COVID-19 years/observations show that the ESG score has a statistically significant negative impact on the ROA, with a coefficient of −0.042 at the 5% significance level. This suggests that higher ESG scores, indicating stronger CSR engagement, might lead to a slight decrease in asset returns, potentially due to the costs or strategic shifts associated with implementing CSR initiatives. Additionally, leverage is found to negatively affect the ROA at the 1% significance level, with a coefficient of −0.006, implying that increased debt relative to equity may reduce the returns generated from assets. The size of the bank, however, does not show a significant impact on the ROA, indicating that the effect of CSR on financial performance is consistent across banks of different sizes.

**Table 12.** Panel Regression Summary Output excluding COVID-19 years/observations. ROA.

|  | Coefficients | Standard Error | *t* Stat | *p*-Value | Lower 95% | Upper 95% | Lower 95.0% | Upper 95.0% |
|---|---|---|---|---|---|---|---|---|
| Intercept | 4.552 | 0.699 | 6.513 | 0.000 | 3.164 | 5.940 | 3.164 | 5.940 |
| ESG | −0.042 ** | 0.017 | −2.512 | 0.014 | −0.075 | −0.009 | −0.075 | −0.009 |
| Size | −0.003 | 0.002 | −1.368 | 0.175 | 0.000 | 0.000 | 0.000 | 0.000 |
| Leverage | −0.006 *** | 0.001 | −4.223 | 0.000 | −0.009 | −0.003 | −0.009 | −0.003 |

** = correlation is significant at the 0.05 level. *** = correlation is significant at the 0.01 level.

In Table 13, regarding the analysis of the return on equity (ROE) excluding COVID-19 years/observations, the regression results reveal a similar pattern in which the ESG score negatively influences the ROE, with a coefficient of −0.273 at the 1% significance level. This significant negative relationship suggests that banks with higher ESG scores might face challenges in maximising equity returns, possibly due to the investment and operational adjustments required for enhanced CSR practices. The negative impact of leverage on the ROE, with a coefficient of −0.038 at the 1% significance level, further supports the notion that higher leverage can constrain profitability and shareholder returns. Similar to the findings for the ROA, the size of the bank does not significantly influence the ROE, highlighting that the observed effects of CSR and leverage on financial performance are not size-dependent. These results underscore the intricate dynamics between CSR engagement, financial leverage, and bank performance, suggesting that while CSR might entail short-term financial trade-offs, it could also reflect broader strategic positioning that influences financial structures and outcomes.

**Table 13.** Panel regression summary output excluding COVID-19 years/observations: ROE.

| | Coefficients | Standard Error | *t* Stat | *p*-Value | Lower 95% | Upper 95% | Lower 95.0% | Upper 95.0% |
|---|---|---|---|---|---|---|---|---|
| Intercept | 31.813 | 4.869 | 6.534 | 0.000 | 22.143 | 41.483 | 22.143 | 41.483 |
| ESG | −0.273 *** | 0.116 | −2.350 | 0.021 | −0.504 | −0.042 | −0.504 | −0.042 |
| Size | −0.002 | 0.002 | −1.433 | 0.155 | 0.000 | 0.000 | 0.000 | 0.000 |
| Leverage | −0.038 *** | 0.010 | −3.827 | 0.0002 | −0.058 | −0.018 | −0.058 | −0.018 |

*** = correlation is significant at the 0.01 level.

To summarise, both the Pearson correlation matrix and the panel regression model produce similar results. In both cases, a negative relationship between the ESG score and the two accounting-based dependent variables, the ROA and ROE, is found. In relation to the control variables in the panel regression model, there is no meaningful correlation between size and the dependent variables, the ROA and ROE, whilst leverage has a negative impact. The same relationships are also found when carrying out the panel regression model and excluding the COVID-19 years/observations.

## 5. Conclusions

The primary objective of this study is to objectively examine the effect of CSR on FP in the UK banking industry for a six-year period from 2017 to 2022.

This study contributes to the vast body of literature on the relationship between CSR and FP, which has produced conflicting results. This research study, as far as our knowledge is concerned, is the first of its sort on UK banks listed on the LSE which collects data from the Bloomberg database. This study empirically examines the relationship between CSR and FP for 32 banks during a six-year period between 2017 and 2022. It discovers that CSR and FP, proxied by two accounting-based metrics, the ROA and ROE, have a negative association. As a result, with a 95% level of confidence, the hypothesis that there is a positive relationship between CSR and FP in UK listed banks is rejected.

In any case, it is crucial to place these negative findings within a larger context. This is not to argue that banks should not invest in CSR initiatives since banks help create a more sustainable global economy in which moral money plays a bigger role. Even though it does not always result in immediate financial rewards, banks are assisting in the creation of a world in which doing the right thing matters. Banks must avoid falling into the trap of performing CSR activities solely for the sake of meeting requirements or greenwashing, when a facade of responsibility is created. Such a check-the-box mentality not only fails to address the core issues at hand but can also undermine the broader credibility and impact of CSR initiatives. A negative CSR-FP relationship challenges the traditional notion of success based solely on FP. The broader societal and environmental impact should be factored into evaluations of a bank's overall performance and contribution to a sustainable economy.

A negative relationship between CSR and FP among UK listed banks has significant policy implications for the broader economic context and regulators and investors, who must navigate the complexities of this relationship to promote responsible banking, foster sustainable economic growth, and align financial success with broader societal well being.

Regulators such as governments, the Financial Conduct Authority (FCA), and the Financial Reporting Authority (FRC) may need to reassess the effectiveness of existing incentive structures that encourage banks to engage in CSR. Policies incentivising CSR activities through regulatory rewards or preferences may require revision to account for the potential negative impact on FP. An example is the green finance strategy, through which the UK government has been actively promoting green finance initiatives (HM Government 2023). The rewards include favourable treatment or incentives for banks that invest in renewable energy projects, support green bonds, or develop sustainable financial products (HM Government 2023). Furthermore, given the potential for greenwashing or superficial CSR efforts, regulators might consider implementing more rigorous disclosure requirements such as mandating the comprehensive and transparent reporting of CSR initiatives.

However, regulators may need to strike a balance between promoting responsible banking practices and safeguarding financial stability. Stricter CSR mandates could influence banks' profitability and risk profiles, requiring regulators to carefully weigh the trade-offs.

Additionally, investors need to recalibrate their investment strategies to consider the negative relationship between CSR and FP. This entails evaluating the long-term value of CSR initiatives, even if they appear to negatively affect immediate financial metrics. Investors should also adopt a more holistic risk assessment approach, recognising that a bank's commitment to CSR might influence its reputation, stakeholder trust, and long-term sustainability, factors that contribute to a bank's overall risk profile.

The following are this study's research limitations. Firstly, increasing the sample size would have led to more conclusive results. However, for research purposes, a six-year sample period allows for a greater focus on the short term and may help establish causal relationships between variables more convincingly because it reduces the influence of confounding factors over time. Secondly, this study was only able to use information from 32 of the 38 listed banks based on data availability in the Bloomberg Database. Thirdly, it is important to acknowledge that the ESG score utilised in this study may not fully encompass the specific CSR initiatives undertaken by a bank due to the multi-dimensional construct of Bloomberg's ESG Disclosure score. In essence, the ESG disclosure activity index employed might not necessarily align with the nature of the actual CSR undertakings pursued by the banks. This limitation is a recurring concern observed in a broader spectrum of CSR-related research. (Nollet et al. 2016; Han et al. 2016; Giannopoulos et al. 2022). Nevertheless, utilising different databases may result in similar conditions (Huang 2022).

This study makes a distinctive contribution to the existing body of literature by introducing UK listed banks into the realm of investigation. For future research proposals, other studies can look at conducting a comprehensive investigation into the influence that external shocks, such as the COVID-19 pandemic and inflationary pressures, have on the relationship between CSR and FP in the UK banking industry. This may show whether excluding specific years characterised by significant disruptions leads to altered outcomes and a reconsideration of the observed negative correlation. Moreover, alternate studies can delve deeper into the multidimensional nature of ESG scores and their relationship to specific CSR actions undertaken by banks. This may explore the alignment or divergence of a bank's ESG disclosure activity index and the actual scope, depth, and impact of its CSR efforts. This could involve qualitative assessments, case studies, or interviews to capture a more in-depth understanding. Further, studies can look at including more UK banks over a longer period to better understand how CSR and FP are connected. This can help us see if the relationship found is true for all banks and if it remains the same over many years.

Each of these proposed avenues for future research has the potential to contribute to a deeper understanding of the complex interplay between CSR and FP within the banking sector and its implications for various stakeholders.

**Author Contributions:** Conceptualization, G.G., N.P. and I.S.; methodology, G.G. and N.P.; software, N.P.; formal analysis, N.P.; writing—original draft preparation, N.P.; writing—review and editing, G.G. and I.S.; supervision, G.G.; project administration, G.G., N.P. and I.S. All authors have read and agreed to the published version of the manuscript.

**Funding:** This research received no external funding.

**Data Availability Statement:** Data are available upon request.

**Conflicts of Interest:** The authors declare no conflict of interest.

## Appendix A

**Table A1.** Sampled Banks.

| Company Name |
| --- |
| AIB GROUP PLC |
| AXIS BANK LIMITED |
| BANCO BILBAO VIZCAYA ARGENTARIA S.A. |
| BANCO SANTANDER S.A. |
| BANKMUSCAT (S.A.O.G.) |
| BANK OF GEORGIA GROUP PLC |
| BANK OF IRELAND GROUP PLC |
| BARCLAYS PLC |
| CLOSE BROTHERS GROUP PLC |
| COMMERCIAL INTERNATIONAL BANK (EGYPT) S.A.E. |
| FEDERAL BANK LIMITED (THE) |
| GUARANTY TRUST HOLDING COMPANY PLC |
| HSBC HOLDINGS PLC |
| INVESTEC PLC |
| JSC HALYK BANK |
| LLOYDS BANKING GROUP PLC |
| METRO BANK PLC |
| NATIONWIDE BUILDING SOCIETY |
| NATWEST GROUP PLC |
| NOVA LJUBLJANSKA BANKA D.D., LJUBLJANA |
| OTP BANK PLC |
| PERMANENT TSB GROUP HOLDINGS PLC |
| SECURE TRUST BANK PLC |
| STANDARD CHARTERED PLC |
| STATE BANK OF INDIA |
| TBC BANK GROUP PLC |
| TCS GROUP HOLDING PLC |
| THE COMMERCIAL BANK OF QATAR (Q.S.C.) |
| TURKIYE IS BANKASI A.S. |
| UNITED BANK LIMITED |
| VIRGIN MONEY UK PLC |
| ZENITH BANK PLC |

Source: (London Stock Exchange 2023).

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
