# Peer review of "What Is the Relationship between Corporate Social Responsibility and Financial Performance in the UK Banking Sector?"

_jrfm, doi:10.3390/jrfm17050187_

Round 1

Reviewer 1 Report

Comments and Suggestions for Authors

Synopsis:

This paper investigates the relationship between CSR and FP in UK-listed banks on the London Stock Exchange (LSE) over a period of six years from 2017 to 2022. Using panel data regression and two proxies for the dependent variable, the results of this study suggest a negative relationship between ESG and both ROA and ROE.

Abstract

The authors are requested to revise the abstract with a greater focus and purpose. Craft an abstract that succinctly covers the main objective of the study, the dataset employed along with its period, the methodology utilized, the significant findings, and ultimately, the implications of the study.

Introduction:

The authors need to articulate their research questions and objectives clearly, identify potential theoretical backgrounds, motivations, or gaps, and explain how their study contributes to the existing literature. What novel insights do they offer regarding the relationship between CSR and FP that have been overlooked by previous research? While numerous studies have explored the association between CSR and FP, it is crucial to distinguish how this study differs from prior work in this area. Merely claiming to be the first study conducted in the UK capital market is insufficient to justify its contribution.

The authors should revise the introduction section, addressing these issues and providing more elaborate discussions on the research motivations and objectives. Additionally, in the introduction, they should succinctly outline the study findings and discuss its theoretical, practical, and policy implications.

Literature review:

The arguments in the Sections 2.2 and 2.3 need more support from the previous studies.

Research method:

Sample Selection

In the "Sample Data" section, the authors need to justify why they chose the study period (2017-2021) and why they conducted the study specifically within the UK context rather than considering all European countries. Further elaboration on these issues is needed.

Data and methodology

The authors are required to reorder this section as follows: 3.1 Sample Data, 3.2 Variable Definitions, and finally 3.3 Model Specifications. In Section 3.3, the authors are required to justify the type of regression used in the study. What is the reason for using a panel data model? Additionally, the authors need to mention whether the regression assumptions are tested or not (e.g., outliers, normality, autocorrelation, Hausman test, etc.).

Empirical results

The authors are required to add Section 4, "Empirical Results," and discuss the descriptive statistics and regression results. Furthermore, in this section, the author(s) are required to elaborate more on the implications or meaning of the figures presented in the descriptive statistics table. What insights do the figures provide, and how do they impact the results? The descriptive statistics should also be linked and used to explain the results or implications of the regression analysis. This is essential for presenting descriptive analysis before conducting inferential analysis in empirical research papers.

Regression results

The authors need to rewrite this section of regression results and amend the presentations of the tables.

I wish the author/s all the best with their study and hope that my comments and suggestions will be useful in taking their paper to the next level.

 Regards

Comments on the Quality of English Language

The authors need to amend the study presentation to align it with the journal format.

Reviewer 2 Report

Comments and Suggestions for Authors

“What is the Relationship between Corporate Social Responsibility and Financial Performance in the UK Banking Sector?” is an interesting article that examines whether there is a relationship between financial performance (using ROE and ROA) and ESG score (which is used as a proxy for Corporate Social Responsibility). 

My only concern is the framing of the paper.  ESG is highly controversial and its overlap with CSR is subjective. I suggest either changing the title to state that the paper is comparing FP and ESG or least addressing this alignment problem in the introduction, so readers know what is actually being compared early.  This is already done in the abstract.

In Section 4.4 (Results Robustness) on page 15, the paper justifies the validity of retaining data from the COVID period stating that the same results are found during the non-COVID period.  If anything, including the COVID period increases the value of the research as it shows performance during a period of financial stress.

Round 2

Reviewer 1 Report

Comments and Suggestions for Authors

I thank the authors for responding to all my previous comments